# How Is Telehealth Currently Being Utilized to Help in Hypertension Management within Primary Healthcare Settings? A Scoping Review

**DOI:** 10.3390/ijerph21010090

**Published:** 2024-01-12

**Authors:** Haerawati Idris, Wahyu Pudji Nugraheni, Tety Rachmawati, Asep Kusnali, Anni Yulianti, Yuni Purwatiningsih, Syarifah Nuraini, Novia Susianti, Debri Rizki Faisal, Hidayat Arifin, Asri Maharani

**Affiliations:** 1Department of Health Administration & Policy, Faculty of Public Health, Sriwijaya University, Indralaya 30662, Indonesia; 2Research Center for Public Health and Nutrition, National Research and Innovation Agency, Central Jakarta 10340, Indonesia; wahy062@brin.go.id (W.P.N.); tetyr272002@gmail.com (T.R.); asep059@brin.go.id (A.K.); anni.yulianti@brin.go.id (A.Y.); yuni033@brin.go.id (Y.P.); syar021@brin.go.id (S.N.); novia.susianti@brin.go.id (N.S.); debr001@brin.go.id (D.R.F.); 3Department of Basic Nursing Care, Faculty of Nursing, Universitas Airlangga, Surabaya 60286, Indonesia; hidayat.arifin@fkp.unair.ac.id; 4Division of Nursing, Midwifery & Social Work, Faculty of Biology, Medicine and Health, The University of Manchester, Manchester M13 9PL, UK; asri.maharani@manchester.ac.uk

**Keywords:** hypertension, primary healthcare, telehealth, telemonitoring

## Abstract

Telehealth has improved patient access to healthcare services and has been shown to have a positive impact in various healthcare settings. In any case, little is understood regarding the utilization of telehealth in hypertension management in primary healthcare (PHC) settings. This study aimed to identify and classify information about the types of interventions and types of telehealth technology in hypertension management in primary healthcare. A scoping review based on PRISMA-ScR was used in this study. We searched for articles in four databases: Pubmed, Scopus, Science Direct, and Embase in English. The selected articles were published in 2013–2023. The data were extracted, categorized, and analyzed using thematic analysis. There were 1142 articles identified and 42 articles included in this study. Regarding the proportions of studies showing varying trends in the last ten years, most studies came from the United States (US) (23.8%), were conducted in urban locations (33.3%), and had a quantitative study approach (69%). Telehealth interventions in hypertension management are dominated by telemonitoring followed by teleconsultation. Asynchronous telehealth is becoming the most widely used technology in managing hypertension in primary care settings. Telehealth in primary care hypertension management involves the use of telecommunications technology to monitor and manage blood pressure and provide medical advice and counselling remotely.

## 1. Background

Most health systems share common goals of improving patient health, being responsive to patient needs, and ensuring financial sustainability [1]. Health systems differ between countries in terms of their structure, financing, and outcomes. Factors such as socioeconomic status, political framework, and cultural diversity contribute to these differences. For example, the US does not have a universal healthcare system, and most Americans receive health insurance through their employers [2]. In contrast, countries like Canada and the UK have universal healthcare systems that are publicly funded [3]. Countries like Japan, Republic of Korea, Singapore, the United Arab Emirates, Australia, and New Zealand provide healthcare to almost all residents, with residents paying some healthcare costs out of pocket [4]. Economic globalization has also played a role, leading to the commercialization of healthcare services and weakening national health systems, particularly in low-income countries [5]. The healthcare spending also varies significantly between countries, with North America spending more than twice as much per capita as the European Union on average [6].

Hypertension has become the major cause of cardiovascular disease and early mortality globally [7]. In 2019, it was found that around 1.28 billion adults aged 30–79 years worldwide had hypertension, with most (two-thirds) living in middle and low-income countries [8]. Elevated blood pressure was responsible for an estimated 4.5 million deaths in men and 4.0 million in women in 2015 [9]. The association between hypertension and premature deaths has also been recorded in several countries [10,11,12].

The increase in the incidence of hypertension is expected to continue [13]. Therefore, hypertension should be detected early and managed properly through education and treatment [14]. This can be managed through the use of telehealth [15]. Telehealth can improve access for isolated people with hypertension in rural areas [16]. Moreover, it might be an acceptable tool by using simple telehealth to diagnose and monitor hypertension among users [17]. In addition, Blood Pressure Telemonitoring is useful both for screening and diagnosing hypertension as well as for improving hypertension management [18]. Likewise, eHealth will support the creation of a network between healthcare professionals in improving screening, hypertension management, and related comorbidities and in the effective prevention of cardiovascular disease [19].

Telehealth technology in developing countries differs from that in developed countries in several ways. Firstly, in developing countries, there is a lack of resources in the health system, leading to challenges in implementing eHealth services [20]. Additionally, the level of development in each country and the commitment of their governments to provide affordable healthcare services play significant roles in determining the success of eHealth models [21]. Furthermore, the asymmetry among healthcare centers, hospitals, and user-ends poses a challenge in fully adopting telehealth technology in developing countries [22]. On the other hand, in developed countries, telehealth services have been slow to be adopted, with uptake being piecemeal and ad hoc [23].

Telehealth can possibly work on the nature of medical services and patient fulfillment. Healthcare providers have widely adopted remote patient monitoring to reduce hospitalization rates and disease management to improve patient self-efficacy [24]. Telehealth can potentially reform and transform the industry by reducing costs and improving quality, access, and patient satisfaction [25,26]. Telehealth is well suited to support patients with chronic, complex, or comorbid conditions, [27] including hypertension. Several previous studies have reported the potential of telehealth to have a positive impact on hypertension management in primary healthcare [28,29,30,31].

The key barriers to hypertension control in primary care include a lack of effective screening and awareness, challenges with accessing treatment, difficulties in managing hypertension once it is treated, medication adherence barriers, lifestyle-related barriers, barriers related to the affordability and accessibility of care, and awareness-related barriers [32,33,34,35]. Patient-related barriers, such as the misinterpretation of blood pressure readings, also contribute to the challenges in hypertension control [36].

Telehealth has shown benefits for hypertension management in primary healthcare. It has the potential to reduce barriers to accessing healthcare and improve clinical outcomes [37]. Telehealth interventions have been used to treat patients with hypertension, heart failure, and stroke, with most interventions employing a team-based care approach [38]. These interventions utilize the expertise of physicians, nurses, pharmacists, and other healthcare professionals to collaborate on patient decisions and provide direct care [39]. Patients using telehealth have seen significant improvements in clinical outcomes such as blood pressure control, which are comparable to patients receiving in-person care [40]. Telehealth can also support team-based care delivery and benefit patients and healthcare professionals by increasing opportunities for communication, engagement, and monitoring outside a clinical setting [41].

Previous literature review studies focused on the benefits and challenges of implementing telehealth in hypertension management in primary healthcare [42,43]. However, research exploring the type of intervention and technology type of telehealth in hypertension management in primary healthcare has not been widely documented. This paper describes the characteristics of telehealth in hypertension management, which can later be used to improve the quality of hypertension services in primary healthcare. The purpose of this review was to identify information on types of interventions and technology from telehealth in the management of hypertension in primary healthcare (PHC) settings.

## 2. Method

The review follows the procedures and recommendations of the Joanna Briggs Institute (JBI) for conducting scoping reviews [44]. The recent JBI guidelines categorize a scoping review as the optimal approach to comprehensively explore the existing literature on a subject by mapping and condensing accessible evidence. Additionally, scoping reviews are well suited to address areas lacking knowledge and offer valuable perspectives to aid decision-making. This evaluation also included PRISMA checklist instructions for reporting an accurate literature review [45]. 

### 2.1. Research Questions

In this study, we identified the characteristics of telehealth in hypertension management in primary healthcare through a scoping review by answering the research questions below:What kind of interventions are carried out in the management of hypertension in PHC settings?What types of technology are used in the management of hypertension in PHC settings?

### 2.2. Research Strategy

The bibliographic databases, such as PubMed, Scopus, Science Direct, and Embase, with the aid of a medical research librarian, were used as a structured literature search in conducting the study. Research questions were developed using a PCC (hypertension, telehealth, and primary healthcare). The search strategy compromised search terms using Medical Subject Headings (MESH) for the concepts “telehealth”, “hypertension”, and “primary healthcare” in the health topic database. We used the Boolean operator AND OR. All references from the databases were exported to Mendeley for duplicating removal and final screening. For a total overview of the applied search, see Table 1. To find the selected article, the authors attempted to obtain full-text versions of the articles using Google Scholar, ResearchGate, and other databases.

### 2.3. Eligible Criteria

All reviewers used the inclusion and exclusion criteria to screen the titles, abstracts, and full articles. The articles we submitted referred to articles in the form of research results published in the last ten years (2013–2023) using the document language English. Articles in the form of scoping reviews, systematic reviews, and literature reviews, not in English and not available in full-text form, were excluded from this study. Publications meeting the inclusive criteria, and those for which the first reviewer (Y.P.) was in doubt, were reviewed a second time by other reviewers (S.N., N.S.). In terms of disagreement, a discussion between all reviewers determined inclusion or exclusion.

### 2.4. Study Selection

We searched and selected papers according to the criteria created and checked for duplicates of existing papers. Three people independently screened them according to title and abstract. The reasons for excluding existing papers until the final results of the selected studies were found are shown in Figure 1.

### 2.5. Data Extraction and Analysis

The data were extracted from chosen documents in full text in the following format: author, year, subject, intervention, type of population, type of technology, telehealth objectives, study design, country, software/hardware tools, and study outcomes. The extraction table is presented in Table 2. The results are presented in descriptive statistics, including frequency and percentage values, using Microsoft Excel 2019. The results of the scoping review of this study are presented in the form of map data on the distribution of telehealth use in various countries using the ArcGIS application. Data are presented in the form of diagrams and tables, according to the aim and scope of the review. Thematic analysis is presented based on the themes found. This procedure includes finding emerging patterns, assigning codes to the data, and combining these codes into broad themes that accurately express information to answer research questions [46].

## 3. Results

A sum of 1442 documents were identified after searching the selected databases. Also, 484 duplicates were removed. After sifting through 958 titles and abstracts, 440 articles were excluded. The remaining 518 articles were checked based on eligibility requirements, and we found 471 bibliography search results assessed for eligibility (including those papers that could not be obtained in full text). The remaining 47 articles were selected. Articles that were not set in primary healthcare and used the wrong document type were excluded in the final selection process, resulting in 42 articles being included in this study. The process of study selection is outlined in Figure 1.

### 3.1. Characteristics of Study

The articles included in this study were articles published between 2013 and 2023. Most of the research came from the United States (USA) (n = 10), followed by Brazil (n = 5) and the UK (n = 3), as shown in Figure 2a. According to the type of research, most of the studies included in this study used a quantitative approach followed by qualitative methods and mixed methods, as presented in Figure 2b. Regarding geographical settings, most studies were conducted in urban locations followed by rural locations, as presented in Figure 2c. The number of studies reporting the use of telehealth in hypertension management tended to fluctuate from 2013 to 2023. The highest numbers of published studies were reported in 2019 and 2022 (n = 7), as presented in Figure 2d.

### 3.2. Type of Intervention

We referred types of telehealth interventions based on the digital intervention categories promoted by WHO to ensure interoperability, i.e., teleconsultation, telemonitoring, teleassistance, and tele-expertise [88]. We added one category telehealth intervention, i.e., tele-education, because some articles used tele-education as an intervention for hypertension management. In this study, we divided types of telehealth interventions into five categories, namely teleconsultation, telemonitoring, teleassistance, tele-expertise, and tele-education. The synthesis of selected paper data using thematic analysis shows that most hypertension management interventions in primary care are of the telemonitoring type, followed by teleconsultation and tele-education interventions, as presented in Table 3.

RQ 1. What are the interventions carried out in the management of hypertension in primary healthcare?

### 3.3. Technology

RQ 2. What is telehealth technology in hypertension management in primary healthcare?

We divided the types of telehealth intervention into three categories based on the study by Mechanic et al. [89], i.e., asynchronous, synchronous, and remote patient monitoring. The results of the data synthesis of selected papers using thematic analysis show that most interventions are of the telemonitoring type, followed by teleconsultation and tele-education interventions, as presented in Table 4.

## 4. Discussion

This study aims to identify information related to the types of interventions and types of telehealth technology in managing hypertension in primary healthcare. There were 42 relevant studies in total that were included in the final synthesis. This study was a scoping review describing the utilization of telehealth in hypertension management in primary healthcare. The articles included in this study were published between 2013 and 2023. Most articles were from the US, followed by Brazil and the UK, most of which were implemented in urban areas. Based on research methods, most articles were analyzed using quantitative methods.

The results of our review show that telehealth management interventions in primary healthcare are dominated by telemonitoring (69%). Telemedicine in hypertension management should include the transmission of vital signs and remote monitoring [90]. In the short-to-medium term, telemonitoring may be more effective than usual care [91]. The implementation of telemonitoring for hypertension can be for routine primary healthcare on a large scale with little impact on physician workload [92]. Telemonitoring can potentially improve the primary care management of Cardiovascular Disease (CVD) by improving patient outcomes and reducing healthcare costs [93]. A literature review also reported that almost all studies reported that telemonitoring was able to reduce blood pressure in the subjects being measured [94].

Telehealth interventions for hypertension management often involve remote patient monitoring (RPM) devices, such as blood pressure monitors, to track patients’ blood pressure levels [95]. These interventions typically employ a team-based care approach, involving physicians, nurses, pharmacists, and other healthcare professionals collaborating on patient decisions and providing direct care [38]. Patients using telehealth for hypertension management have seen significant improvements in clinical outcomes, including blood pressure control, which are comparable to patients receiving in-person care [40].

Apart from telemonitoring, our findings also found teleconsultation to be a frequently used intervention in hypertension management in primary care. Teleconsultation is an effective alternative to face-to-face consultation for many patients presenting to primary healthcare [96]. Teleconsultations between non-physicians and doctors located remotely have the potential to reduce the number of referrals to central clinics [97,98]. Teleconsulting services improve the compatibility of primary services and integration with secondary services in rural communities [99].

Our findings show that hypertension management technology in primary healthcare predominantly uses asynchronous technology. Most recent generation telemedicine systems use an asynchronous telemedicine approach [100]. This technology is reliable, simple to operate, has consistent connections, uses standard communication protocols, and has efficient bandwidth. For example, eHealth and mHealth are gradually gaining key roles in managing hypertensive patients [19]. Additionally, Java client applications that send digital camera images and structured XML text as e-mails are designed for use in resource-poor, poorly networked developing countries [101]. The application of advanced technology in rural healthcare settings has the potential to lower the cost of patient care and managed care insurance plans, enabling expert consultation from remote centers [102]. Asynchronous telehealth can shorten waiting times, reduce unnecessary referrals, and increase patient and provider satisfaction levels [103].

Our findings show that telehealth is currently being utilized to help manage hypertension in primary healthcare. It also helps to provide remote access to healthcare professionals and enables the monitoring of blood pressure [37,95]. It has been increasingly used during the COVID-19 pandemic to ensure continuity of care and improve access to healthcare services [104]. Studies have shown that telehealth interventions, including remote patient monitoring (RPM) and team-based care, have been effective in treating patients with hypertension and cardiovascular disease (CVD) [37]. A collaborative nephrologist–pharmacist telehealth clinic has also been successful in improving difficult-to-control hypertension in patients with chronic kidney disease (CKD) [105]. Telehealth-delivered approaches, such as the TEAM intervention, have been shown to improve hypertension care delivery and blood pressure control [39]. The COVID-19 pandemic has further accelerated the use of telehealth for the management of hypertension and other non-communicable diseases (NCDs) [39]. Overall, telehealth has the potential to reduce barriers to accessing healthcare, improve clinical outcomes, and extend services to remote areas, making it a valuable tool in the management of hypertension in primary healthcare settings.

Health systems and healthcare differ between countries. Despite these differences, most health systems aim to improve patient health, be responsive to patient needs, and ensure financial sustainability [106]. Developing countries face more challenges in building strong and reliable health systems compared to developed countries, leading to disparities in public health status and health problems [3]. Health inequalities exist within and between countries, with socioeconomic and cultural inequalities driving health inequities [106].

Our findings show that different countries use different technologies for hypertension management. Telehealth and eHealth technology availability differ between developed and developing countries. In developed countries, the use of telehealth and eHealth services has been increasing, especially during the COVID-19 pandemic [107]. These countries have reached thresholds for telehealth provision, such as a certain level of telecommunication accessibility, a proportion of elders exceeding 10%, or a proportion of health spending occupying more than 3–5% of GDP [108]. On the other hand, developing countries are also utilizing telehealth and eHealth technology, but the models and approaches vary depending on the level of development and government commitment to providing affordable healthcare services [109].

### Strengths and Limitations

This study captured the use of telehealth for hypertension management in primary healthcare. The selection process uses four databases with systematic procedures. With wide location coverage, it can describe the implementation of telehealth in various countries. One limitation of this study was that it did not include grey literature reviews in the inclusion criteria. Another limitation was language bias because we searched for articles in English only. Future studies can overcome this limitation.

## 5. Conclusions

Telehealth is considered to have the potential for the management of hypertension in primary care. The findings of this study show that telehealth interventions in hypertension management in primary healthcare are dominated by telemonitoring. The technology used is more common in asynchronous telehealth. Further studies are needed to evaluate telehealth services in supporting the management of hypertension in primary care.

## Figures and Tables

**Figure 1 ijerph-21-00090-f001:**
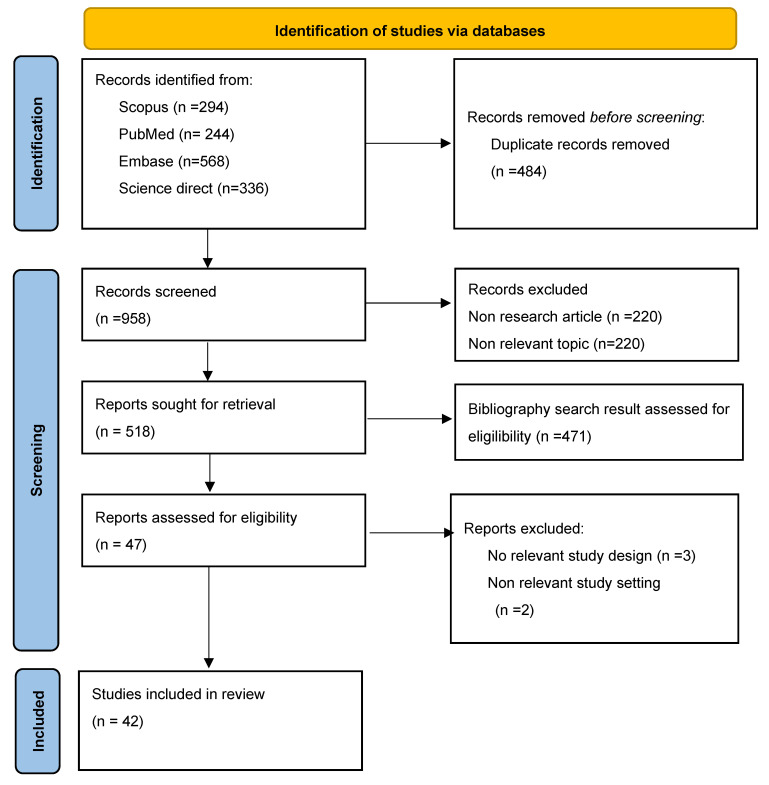
Flowchart of the PRISMA scoping review study identification and selection process.

**Figure 2 ijerph-21-00090-f002:**
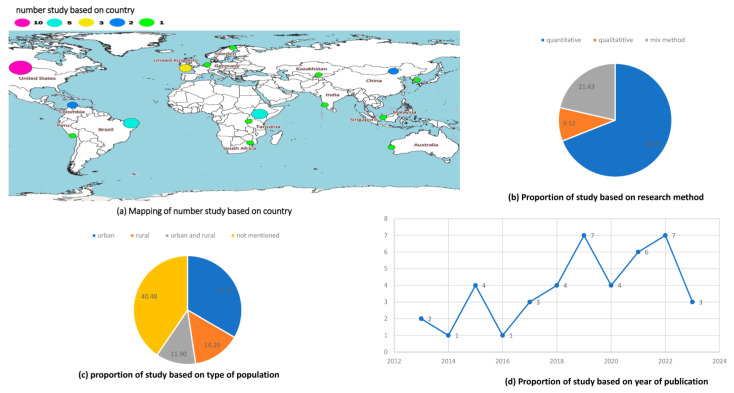
Characteristics of telehealth in hypertension management in primary care settings.

**Table 1 ijerph-21-00090-t001:** Search strategy for databases.

Databases	Keywords and Query
PubMed	“Telemedicine” [Mesh] OR “Mobile health” [tw] OR mHealth [tw] OR eHealth [tw] OR “Tele-Referral” [tw] OR Tele Referral [tw] Filters: in the last 10 years “Hypertension” [Mesh] OR “hypertensi” [tw] OR “high blood pressure” [tw] Filters: in the last 10 years “Primary Health Care” [Mesh] OR “primary health care” [tw] OR “primary care” [tw] Filters: in the last 10 years
Scopus	(TITLE-ABS-KEY (telemedicine) or title-abs-key (“mobile health”) or title-abs-key (mhealth) or title-abs-key (ehealth) or title-abs-key (tele referral) and title-abs-key (hypertension) or title-abs-key (“high blood pressure”) and title-abs-key (“primary health care”) or title-abs-key (“primary care”)) and pubyear > 2012 and pubyear > 2012
Science direct	“high blood pressure” AND telemedicine OR “mobile health” OR “eHealth” AND “primary health care”
Embase	((‘high blood pressure’/exp OR ‘high blood pressure’ OR ‘hypertension’/exp OR hypertension) AND (‘telemedicine’/exp OR telemedicine) OR ‘mobile health’/exp OR ‘mobile health’ OR ‘ehealth’/exp OR ehealth) AND (‘primary health care’/exp OR ‘primary health care’) AND ((controlled clinical trial)/lim OR (randomized controlled trial)/lim) AND (2013–2023)/py

**Table 2 ijerph-21-00090-t002:** Data extraction results.

Author	Year	Participant Subject	Intervention	Population Type	Type of Technology	Purpose of Telehealth	Type of Study	Number Primary Healthcare	Country	Tool of Hardware/Software	Outcome
Barsky et al. [47]	2019	Canadian Aboriginal and Tanzanian communities	SMS-text-messaging-based system for blood pressure measurement and hypertension management	rural	Mobile health (SMS text messaging)	Monitoring blood pressure	mixed methods	n/a	Canada and East Africa	wireless, Bluetooth	Difference in blood pressure reduction for active hypertension management messages or passive health behavior messages Quantitative data on blood pressure reading transmissions Qualitative data collected on the operational aspects of the system from healthcare providers, participants, and community leadership
Naqvi et al. [48]	2022	Acute stroke patients with hypertension	TASC (Telehealth After Stroke Care)	Urban	home blood pressure telemonitoring	Monitoring blood pressure	Pilot randomized trial	n/a	Northern Manhattan	Tablet and monitor	Adherence to video visits Retention Home systolic BP Systolic BP control
Vedanthan et al. [49]	2015	nurses	Tablet-based Decision Support and Integrated Record keeping (DESIRE) tool	rural	Mobile health (mHealth)	management of hypertension	investigative study	n/a	Rural Western Kenya	tablet	Usability of the DESIRE tool in the setting of nurse management of hypertension in Rural Western Kenya Identification of critical incidents Suggested design changes
Dos Santos et al. [50]	2013	professionals and hypertensive patients	Education program	rural	Web conference	Increase the adherence to the treatment of hypertension.	before–after study	2	Brazil	n/a	Adherence to antihypertensive drugs Adherence to low-salt diet Adherence to physical activity
Buis et al. [51]	2020	people with hypertension, medical assistants, physicians, a nurse, and the current and former director of the Family Medicine clinic	BPTrack	urban and rural	Mobile health (mHealth)	Home blood pressure monitoring	pre-post pilot study	1	USA	mobile applications	Change in blood pressure (primary outcome) Change in medication adherence (secondary outcome)
Koopman et al. [52]	2014	patients, nurses, and physicians	Home blood pressure telemonitoring	n/a	Electronic medical record and home blood pressure telemonitoring	blood pressure monitoring	qualitative study	6	South America	USB computer connection, dedicated telemonitoring device with an analog phone line	Blood glucose levels Blood pressure levels
Parker et al. [53]	2018	hypertension patients	text based telemonitoring system	n/a	Home blood pressure telemonitoring	blood pressure monitoring	prospective cohort study	37	South-East Scotland	automatic-transmission system	Occurrences of systolic and diastolic blood pressure Preference for systolic 134 and diastolic 84 (the threshold for alerts was 135/85)
Fisher et al. [54]	2019	hypertension patients	A home-based BP control program	n/a	telemonitoring	blood pressure monitoring	prospective cohort implementation	n/a	USA	home monitors	Blood pressure control rates
Ma et al. [55]	2022	Chinese hypertensive patients	Smartphone-enhanced nurse-facilitated self-care intervention	urban	mobile health	hypertension management	Randomized controlled trial with a repeated-measures design	2	China	smartphone	Body weight Body mass index Diastolic blood pressure Waist circumference Self-care behavior Self-care motivation Self-care self-efficacy Systolic blood pressure
Levine et al. [56]	2018	primary care patients with hypertension	virtual visits	n/a	asynchronous online	hypertension management	propensity-score-matched, retrospective cohort study with adjustment by difference in differences	n/a	USA	n/a	Adjusted difference in mean systolic blood pressure Emergency department visits Specialist office visits Primary care office visits Inpatient admissions
Ashjian et al. [57]	2019	hypertensive patients	an interactive voice response (IVR)	n/a	Electronic health record	home blood pressure monitoring	observational study	14	USA	Aspect Patient Engagement Solution and Microsoft Dynamics 365 platforms	IVR call completion rates Number of hypertensive patients enrolled
De Luca et al. [58]	2021	patients diagnosed with hypertension and professionals	integrated management hypertension	n/a	digitally enabled integrated approach (HER), smartphone, computer	hypertension management	user-centered approach	n/a	Europe	n/a	Functional requirements Use cases
Chen et al. [59]	2023	individual	online health management	urban	internet based	hypertension management	longitudinal study	n/a	China	n/a	Amount of exercise Diastolic blood pressure High-density lipoprotein cholesterol levels Smoking cessation rate Low-density lipoprotein cholesterol levels Systolic blood pressure
Jindal et al. [60]	2018	people with hypertension and diabetes along with comorbid conditions	Smartphone application (mWellcare)	rural	mobile health	integrated management of hypertension	n/a	5	India	tablet-computer-based application	Acceptability Feasibility
Doocy et al. [61]	2017	people aged 40 years or older with hypertension	Mobile health app	urban	mobile health	improve adherence to guidelines and quality of care	a longitudinal cohort study	10	Lebanon	tablets	Adherence to medication Recording of Blood Pressure (BP) readings Recording of blood sugar measurements Recording of Body Mass Index (BMI) reporting Proportion of patients for whom blood sugar, BP, weight, height, and BMI were recorded using the tablet compared with in-paper records Proportion of providers offering lifestyle counseling Provider inquiry of medical history Patient report of provider inquiry regarding medication complications
Leon et al. [62]	2015	female and male participants in South Africa aged 36 to 78 years old	SMS texts	n/a	mobile health	improve adherence to clinic visits and treatment	an individually randomized controlled trial	1	South Africa	Mobile phone	Blood pressure control
Buis et al. [63]	2017	African American patients with uncontrolled hypertension	automated text message	urban	mobile health	improve medication adherence among African Americans with uncontrolled HTN	unblinded randomized controlled pilot trials	2	USA	n/a	Diastolic Blood Pressure (DBP) Medication adherence Systolic Blood Pressure (SBP) Medication adherence self-efficacy Participant satisfaction
Cottrell et al. [17]	2015	patients and clinicians in a national primary care population in England	Text messaging (‘Florence’)	n/a	mobile health	for diagnosis and management hypertension	evaluation study	n/a	UK	Mobile phone	Ascertainment of a diagnosis of hypertension Achievement of hypertension control
Ju et al. [64]	2022	Patients aged ≥19 years were diagnosed with hypertension, diabetes, dyslipidemia, or metabolic syndrome	mobile self-management healthcare app	n/a	Mobile health	management of chronic conditions	pilot multicenter real world study	17	Republic of Korea	mobile app	Anxiety Body weight Sleep quality Sleep duration Quality of life Depression Stress BMI Waist circumference Blood sugar levels Blood pressure Blood lipid levels
Nurakysh et al. [65]	2022	patients with diagnosed arterial hypertension	Mobile application “MyTherapy”	n/a	mobile health	evaluation of the degree of adherence of patients determined to have hypertension to treatment	a multicenter randomized controlled study	1	Kazakhstan	mobile phone app	Adherence to antihypertensive treatment
Manusov et al. [66]	2019	people with chronic illness, obesity, hypertension, hypercholesterolemia, hypertriglyceridemia, and depression	UniMóvil, a mobile health clinic	rural	mobile health	improve poor healthcare access delivery	a retrospective review of the cohort	1	USA	n/a	Obesity Diabetes Hypertension Hypertriglyceridemia Low High Density Lipoprotein Cholesterol (HDL C) Levels Depression Health Related Quality of Life (HRQOL) Domains
Lee et al. [67]	2022	people aged 18 to 75 years, predominantly female, within the University of Pennsylvania Health Systems	remote blood pressure monitoring	urban	Electronic health record	remote blood pressure monitoring	cohort study	n/a	USA	n/a	Number of EHR alerts for persistently elevated BP home readings Number of alerts that clinicians acted on Percentage of alerts that clinicians acted on Type of management used by clinicians (remote or office based) Number of alerts that did not result in changes to the care plan Reasons for not changing the care plan
Marcolinoet al. [68]	2021	people in Brazil, 71% of which were female, consisting of physicians and nurses	teleconsultation	urban and rural	Asynchronous	hypertension management	mixed methods	34	Brazil	Web-based	Perceived feasibility Usability Utility Satisfaction
Peters et al. [69]	2017	hypertensive patients aged above 18 years	phone call and short-message-service text messaging	n/a	mobile health	blood pressure control	quality improvement study	1	USA	Mobile phone	Mean Systolic Blood Pressures (SBPS) Mean Diastolic Blood Pressures (DBPS)
Debon et al. [70]	2020	female humans in Brazil with arterial hypertension who were workers or retirees	use of a mobile health app	n/a	mobile health	monitoring patients with arterial hypertension (AH)	a non-randomized, controlled, non-blind trial	n/a	Brazil	smartphone	Systolic and diastolic Blood Pressure Food Frequency Questionnaire Appraisal Of Self Care Agency Scale Hemogram Creatinine Uric Acid Sodium Potassium Lipid Profile Glycemia
Davoudi et al. [71]	2020	adults with poorly controlled hypertension	an automated text messaging	n/a	mobile health	hypertension management	secondary analysis of data from a randomized controlled trial	1	USA	n/a	Unique phenotypes of patient interactions with an automated text messaging platform for BP monitoring Association between interaction phenotypes and achieving the target BP
McManuset al. [72]	2021	people with treated but poorly controlled hypertension (>140/90 mm Hg) and access to the internet	Home and Online Management	n/a	Home blood pressure telemonitoring	hypertension management	randomized controlled trial	76	UK	Omron	Difference in systolic blood pressure (Mean of 2nd and 1/3 readings) after 1 year Difference in diastolic blood pressure after 1 year
Chew et al. [73]	2023	patients and clinical staff	a remote blood pressure monitoring program	urban	Home blood pressure telemonitoring	blood pressure monitoring	a secondary qualitative study	n/a	Singapura	a Bluetooth-enabled device	A better patient–provider partnership based on the mutually trusted data Patients felt reassured Patients trusted the telehealth program Staff felt that the data were trustworthy Patients’ distrust of technology Clinicians’ concerns about the limitations of technologically mediated interactions Uncertainty experiences
Anderssonet al. [74]	2021	patients and 15 professionals	interactive web-based system	Urban and rural	mobile health	strengthening patients’ potential for self-management	qualitative substudy of a randomized controlled trial	n/a	Sweden	Mobile phone	Partnership between patients and healthcare professionals Documentation of BP treatment Roles of patients and professionals in hypertension management Knowledge of BP values and their relation to daily activities and treatment Patients’ understanding of hypertension Communication regarding BP and lifestyle
Kassavou et al. [75]	2019	healthcare providers, commissioners, and patients with either hypertension or both hypertension and type 2 diabetes	highly tailored text and voice message	n/a	mobile health	to increase adherence to medication in primary care	descriptive and interventional study	n/a	UK—England	Mobile phone	Adherence to medication Intervention engagement Fidelity of intervention content Awareness about the necessity to take and maintain adherence to medication Reinforced social support and habit formation Reminded patients to take medication as prescribed
Cottrell et al. [76]	2015	patients and professional users in England with hypertension, CKD, and diabetes	text messages	n/a	mobile health	support self-management and education using technology with which patients are already familiar	evaluation study	425	UK	Mobile phone	Agreement with the adapted friends and family statement Professional and patient user satisfaction with the aim for health program Patient activity using florence Minimum target days of texting for hypertension protocols Patient responses to evaluative texts Professional user satisfaction with the program
Abdullah et al. [77]	2016	patients with hypertension and comorbidities	a blood pressure telemonitoring service	urban	home BP telemonitoring	blood pressure monitoring	a qualitative study design	n/a	Malaysia	MediHome Digital Blood Pressure and Pulse Oximeter 2-in-1 Monitor	Patients’ acceptance of a BP telemonitoring service delivered in primary care Patients’ views and experiences of the BP telemonitoring service Patients’ struggles with the perceived usefulness of the BP telemonitoring service Patients’ confusion in making sense of the monitored home BP readings Patients’ feedback on the BP telemonitoring functionalities to improve interactions Patients’ thoughts about the implications of the monitored home BP readings for their hypertension management and overall health
Nau et al. [78]	2021	patients aged 40–70 years	videos, web-based education, and text message	urban	mobile health	To support patients with improving lifestyle behaviors for high blood pressure	pilot study	n/a	Australia	Mobile phone	Intervention feasibility and acceptability Patient ratings of content and behavior changes Changes in behaviors
Ye et al. [79]	2022	patients with hypertension aged between 18 and 85 years	video and telephone	urban	telemedicine visit	controlling high blood pressure	retrospective cohort study	n/a	South America	n/a	Failure to meet the controlling-high-blood-pressure quality measure Poor BP control
Calderón et al. [80]	2023	patient with hypertension	SMS-based home BP telemonitoring system	urban	home bp telemonitoring	helps improve adherence to treatment, also improving disease awareness	randomized controlled trial	1	Peru	omron	Change in systolic BP values Change in diastolic BP values
Sin et al. [81]	2020	People aged 21–70 years old in Singapore with Type 2 DM and/or hypertension	Telemonitoring	urban	telemonitoring	diabetes and hypertension management	cross-sectional survey	2	Singapura	n/a	Willingness of patients with type 2 Diabetes Mellitus (T2Dm) and/or hypertension towards the utilization of telemonitoring
Cimini et al. [82]	2022	primary care physicians, one nurse, one pharmacist, and one community health worker	a digital solution with a decision support system (DSS) for community health workers (CHWs)	n/a	telemedicine with video consultations	To address and identify at risk patients with uncontrolled hypertension or diabetes mellitus (DM)	multimethodological	34	Brazil	video consultation	Frequency of consultations Control of hypertension Control of diabetes mellitus Systolic blood pressure Diastolic blood pressure Glycohemoglobin levels
Shaw et al. [83]	2013	US stakeholders including physicians, nurses, non-physician providers, administrators, and an IT professional with hypertension	nurse-delivered self-management phone	urban and rural	telephone counselling	initiating and maintaining specific health behaviors related to hypertension	mixed methods approach	3	USA	n/a	Level of organizational readiness to implement the intervention Specific barriers, facilitators, and contextual factors that may affect organizational readiness to change Organizational barriers and facilitators of successful implementation
Grant et al. [84]	2019	people in the West Midlands, UK with hypertension, including patients, healthcare professionals, and patient caregivers	Text message	urban and rural	mobile health	blood pressure monitoring	randomized controlled trial	n/a	UK	mobile platform	Acceptability of self- and telemonitoring to patients and HCPS Communication Managing data Integrating self-monitoring into hypertension management (structured care)
Saleh et al. [85]	2018	Lebanese hypertensive	Short message service (SMS)	rural	mobile health	enhance access among underserved rural and refugee populations to health services specific to hypertension and/or diabetes.	mixed methods	n/a	Lebanon	Mobile phone	Perceived usefulness of SMSs Perceived ease of reading and understanding SMSs Compliance with SMSs through daily behavioral modifications Receiving and not reading SMSs Behavior change across settings
Vitório et al. [86]	2019	hypertensive patients	TeleHAS (tele-hipertensão arterial sistêmica, or arterial hypertension system)	urban	computerized clinical decision support system (CDSS)	hypertension management	Mixed methods	88	Brazil	n/a	Feasibility Usability Utility
Teo et al. [87]	2021	middle-aged people in Asia with hypertension	Home blood pressure monitoring	urban	Home blood pressure monitoring, teleconsultation	hypertension management	a mixed-methods field study	n/a	Singapura	Bluetooth	Blood pressure control Cost-effectiveness Patient satisfaction

**Table 3 ijerph-21-00090-t003:** Thematic synthesis based on type of intervention.

Key Themes	References
Tele-consultation	Dos et al. [50], Fisher et al. [54], Ma et al. [55], De Luca et al. [58], Chen et al. [59], Jindal et al. [60], Leon et al. [62], Buis et al. [63], Ju et al. [64], Nurakysh et al. [65], Manusov et al. [66], Marcolino et al. [68], Chew et al. [73], Andersson et al. [74], Kassavou et al. [75], Shaw et al. [83], Vitório et al. [86], Teo et al. [87]
Tele-monitoring	Barsky et al. [47], Naqvi et al. [48], Vedanthan et al. [49], Buis et al. [51], Koopman et al. [52], Parker et al. [53], Fisher et al. [54], Levine et al. [56], Ashjian et al. [57], De et al. [58], Doocy et al. [61], Ju et al. [64], Nurakysh et al. [65], Lee et al. [67], Marcolino et al. [68], Peters et al. [69], Debon et al. [70], Davoudi et al. [71], McManus et al. [72], Chew et al. [73], Andersson et al. [74], Cottrell et al. [76], Abdullah et al. [77], Ye et al. [79], Calderón et al. [80], Sin et al. [81], Cimini et al. [82], Grant et al. [84], Vitório et al. [86]
Tele-expertise	Fisher [54], Jindal [60], Marcolino [68], Vitório [86]
Tele-assistance	Dos et al. [50]
Others: tele-education	Dos et al. [50], Ma et al. [55], De et al. [58], Manusov et al. [66], Marcolino et al. [68], Cottrell et al. [76], Nau et al. [78], Shaw et al. [83], Saleh et al. [85], Vitório et al. [86]

**Table 4 ijerph-21-00090-t004:** Thematic synthesis based on the type of technology.

Key Themes	References
Asynchronous: SMS text messaging, Tablet-based Decision Support and Integrated Record keeping (DESIRE) tool (mobile health), mobile health app, smartphone, patient-texted system, SMS and IVR messages, internet-based health management, mobile health clinic, mobile health, clinical decision support system, clinical decision support system, web conference	Cottrell et al. [17], Barsky et al. [47], Vedanthan et al. [49], Dos et al. [50], Buis et al. [51], Parke et al. [53], Ma et al. [55], Chen et al. [59], Jindal et al. [60], Doocy et al. [61], Leon et al. [62], Buis et al. [63], Ju et al. [64], Manusov et al. [66], Marcolino et al. [68], Peters et al. [69], Debon et al. [70], Davoudi et al. [71], Kassavou et al. [75], Cottrell et al. [76], Nau et al. [78], Grant et al. [84], Saleh et al. [85], Vitório et al. [86]
Synchronous: Virtual visit, teleconsultation online, interactive voice response (IVR), Home and Online Management and Evaluation of Blood Pressure, web-based system, telemedicine visit, telephone, app with decision support system	Naqvi et al. [48], Levine et al. [56], Ashjian et al. [57], McManus et al. [72], Andersson et al. [74], Ye et al. [79], Cimini et al. [82], Shaw et al. [83]
Remote patient monitoring: Blood Pressure Remote Patient Monitoring (RPM), monitoring hypertension, telemonitoring, home BP telemonitoring system	Koopman et al. [52], Fisher et al. [54], De Luca et al. [58], Lee et al. [67], Chew et al. [73], Abdullah et al. [77], Calderón et al. [80], Sin et al. [81], Teo et al. [87]

## Data Availability

The data presented in this study are available on request from the corresponding author.

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
