# Peer review of "How Is Telehealth Currently Being Utilized to Help in Hypertension Management within Primary Healthcare Settings? A Scoping Review"

_ijerph, 2024, doi:10.3390/ijerph21010090_

Round 1
Reviewer 1 Report
Comments and Suggestions for Authors
A manuscript with a very interesting title, but the work requires solid refinement. The abstract requires rewording. It contains sentences that are not substantive and do not add much to the value of the work. These are the following sentences:
Telehealth has the potential to improve the quality of patient and health care satisfaction.
The results of this scoping review show the characteristics of telehealth in hypertension management in primary care settings.
I recommend that the authors refine the abstract. They started by justifying the importance of the topic, stating what the main purpose of the manuscript was, how, when, and on what sample the research was conducted, and what the result of the work was. This information is missing.
Sentence line 55-57 is not clear: This
scoping review goals to identify information on types of interventions and technology from telehealth in the management of hypertensionin primary healthcare (PHC) settings." there is no information in how many primary healthcare settings the study was conducted and in which country it was located. The paper should characterize the issue of "primary healthcare"
The authors write about 5 categories in a sentence and mention only 4: "We divided types of telehealth interventions into five categories based on the digital intervention categories promoted by WHO to ensure interoperability, i.e., teleconsultation, telemonitoring, teleassistance, and teleexpertise"
On line 166, the authors use the abbreviation CVD without explanation.
The discussion section is definitely too short and does not refer to the results achieved. In this section, authors should answer the question posed in the title:
How is telehealth currently being utilized to help in the hypertension’s management within primary healthcare settings?
This reply did not appear. Also in this section, there should be a reference to the achievements of the literature on the topic, of which there is also none.
The work contains minor grammatical and punctuation errors.
Comments on the Quality of English LanguageThe work contains minor grammatical and punctuation errors.
Author Response
Response to Reviewer 1 Comments
|
Summary Thank you for reviewing our paper. We are grateful for your insightful comments and helpful changes to our paper. We have addressed the reviewers' concerns, considered their suggestions, and made the necessary adjustments. Questions for General Evaluation Reviewer’s Evaluation Is the work a significant contribution to the field? |
|
|
Is the work well organized and comprehensively described? |
|
|
Is the work scientifically sound and not misleading? |
|
|
Are there appropriate references to related and previous work? |
|
|
Is the English used correct and readable? |
|
|
Response and revisions: Thank you for your feedback on this study. We appreciate it and are trying to improve the quality of this study. We are also adding appropriate references to related and previous work. Based on the reviewer's suggestion, we have reorganised and described the study in more detail. We have addressed the reviewers' concerns, considered their suggestions, and made the necessary adjustments. Below the detailed responses to reviewer comments and concerns. We have indicated on which page and line the correction has been done. |
Point-by-point response to Comments and Suggestions for Authors
Comments 1: A manuscript with a very interesting title, but the work requires solid refinement. The abstract requires rewording. It contains sentences that are not substantive and do not add much to the value of the work. These are the following sentences:
Telehealth has the potential to improve the quality of patient and health care satisfaction.
The results of this scoping review show the characteristics of telehealth in hypertension management in primary care settings.
Response 1: Thank you for raising this issue.
We have revised our abstract in lines 4-5 and 18-20
Telehealth has improved patient access to healthcare services and has been shown to have a positive impact in various healthcare settings
Telehealth in primary care hypertension management involves the use of telecommunications technology to remotely monitor and manage blood pressure, provide medical advice and counselling.
Comments 2: I recommend that the authors refine the abstract. They started by justifying the importance of the topic, stating what the main purpose of the manuscript was, how, when, and on what sample the research was conducted, and what the result of the work was. This information is missing.
Response 2: We appreciate this comment and suggestion.
We have revised our abstract following the suggestion (lines 4-20).
Comments 3: Sentence line 55-57 is not clear: This
Response 3: Thank you for the comment.
We have revised the sentences in lines 93-95 by changing the word “this”.
The purpose of this review was to identify information on types of interventions and technology from telehealth in the management of hypertension in primary healthcare (PHC) settings.
Comments 4: scoping review goals to identify information on types of interventions and technology from telehealth in the management of hypertension in primary healthcare (PHC) settings." there is no information in how many primary healthcare settings the study was conducted and in which country it was located. The paper should characterize the issue of "primary healthcare"
Response 4: Thank you for these constructive comments.
We have added information on the number of primary health care facilities where the study was conducted and the country where it occurred (Table 3).
Furthermore, we have added information about the issue of primary healthcare in the introduction in lines 71-87.
Comments 5: The authors write about 5 categories in a sentence and mention only 4: "We divided types of telehealth interventions into five categories based on the digital intervention categories promoted by WHO to ensure interoperability, i.e., teleconsultation, telemonitoring, teleassistance, and teleexpertise"
Response 5: Thank you for raising this point.
We have revised the sentence and added an explanation about the type of telehealth intervention categories in lines 2-7.
We referred to types of telehealth interventions based on the digital intervention categories promoted by WHO to ensure interoperability, i.e., teleconsultation, telemonitoring, teleassistance, and teleexpertise. We added one category, telehealth intervention, i.e., tele-education, because some articles use tele-education as an intervention for hypertension management. In this study, we divided types of telehealth interventions into five categories: teleconsultation, telemonitoring, teleassistance, teleexpertise and tele-education.
Comments 6: On line 166, the authors use the abbreviation CVD without explanation.
Response 6: Thank you for the suggestion.
We have added the full form of CVD in line 43.
Comments 7: The discussion section is definitely too short and does not refer to the results achieved. In this section, authors should answer the question posed in the title:
How is telehealth currently being utilized to help in the hypertension’s management within primary healthcare settings?
This reply did not appear. Also in this section, there should be a reference to the achievements of the literature on the topic, of which there is also none.
Response 7: Thank you for raising this point.
We have revised our discussion and added more information about the meaning, importance, and relevance of our findings to answer the question posed in the title:
How is telehealth currently being utilized to help in the hypertension’s management within primary healthcare settings. Furthermore, we have added the references related to this part.
See lines 60-85
Comments 8: Comments on the Quality of English Language
The work contains minor grammatical and punctuation errors.
Response 8: Thank you for raising this point.
We have revised the minor grammatical and punctuation errors in this manuscript.

Reviewer 2 Report
Comments and Suggestions for Authors
This is a very interesting study focused on describing the characteristics of telehealth in hypertension management and identifying information on types of interventions and technology from telehealth in the management of hypertension in primary healthcare (PHC) settings. It has some flaws that, in my opinion, should be addressed before being published.
Introduction
Originality and justification. OK.
Please add background on expected differences between countries and health systems. It is not the same as private vs public primary care. And telehealth /e-health technology in developing vs developed countries.
Please correct typos, line 36 says “hypertion” I suppose it should be “hypertension”; and line 57 says “hypertensionin” I suppose it should be “hypertension in”-
Material and methods. Ok
Results. Table 3:
Ref Barsky et al.27, Country, should say Canada in addition to East Africa
Ref Koopman et al. 32 , revise the name of the country.
Ref Fisher et al. 34, revise the name of the country.
Ref Levine et al. 36, revise the name of the country.
Ref Ashjian et al. 37, revise the name of the country.
Ref Manusov et al. 46, revise the name of the country.
Ref Lee et al. 47, revise the name of the country.
Ref Davoudi et al. 51, revise the name of the country.
Ref Nau et al. 58, revise the name of the country.
Ref Ye et al. 59, revise the name of the country.
Discussion. Please comment on the differences between countries and health systems. It is not the same as private vs public primary care. And telehealth /e-health technology in developing vs developed countries.
Limitation. Authors should recognize publication bias, because of the language. They only included articles in English.
Comments on the Quality of English LanguagePlease correct typos, line 36 says “hypertion” I suppose it should say “hypertension”; and line 57 says “hypertensionin” I suppose it should say “hypertension in”.
Author Response
Response to Reviewer 2 Comments
Summary
Thank you for reviewing our paper. We are grateful for your insightful comments and helpful changes to our paper. We have addressed the reviewers' concerns, considered their suggestions, and made the necessary adjustments.
|
1. Questions for General Evaluation |
Reviewer’s Evaluation |
|
Is the work a significant contribution to the field? |
thhhghhjgkjh |
|
Is the work well organized and comprehensively described? |
|
|
Is the work scientifically sound and not misleading? |
|
|
Are there appropriate references to related and previous work? |
|
|
Is the English used correct and readable?
Response and revisions: Thank you for your feedback on this study. We appreciate it and are trying to improve the quality of this study. We have reorganised and described it in more detail based on the reviewer’s suggestion. |
Point-by-point response to Comments and Suggestions for Authors
Comments 1: This is a very interesting study focused on describing the characteristics of telehealth in hypertension management and identifying information on types of interventions and technology from telehealth in the management of hypertension in primary healthcare (PHC) settings. It has some flaws that, in my opinion, should be addressed before being published.
Response 1: Thank you for this comment and suggestion.
We have addressed the reviewers’ concerns, considered their suggestions, and made the necessary adjustments. Below are the detailed responses to reviewer comments and concerns, which include the information on the page and the line of the revisions.
Comments 2: Introduction
Originality and justification. OK.
Response 2: Thank you for your feedback on this section.
Comments 3: Please add background on expected differences between countries and health systems. It is not the same as private vs public primary care. And telehealth /e-health technology in developing vs developed countries.
Response 3: Thank you for drawing our attention to this important issue.
We have added some information in the background about differences between countries and health systems (lines 25-37) and telehealth /e-health technology in developing vs developed countries (lines 54-62).
Comments 4: Please correct typos, line 36 says “hypertion” I suppose it should be “hypertension”; and line 57 says “hypertensionin” I suppose it should be “hypertension in”-
Response 4: Thank you for the correction. We have changed the word "hypertion" to "hypertension" (line 48) and the word "hypertensionin" to "hypertension in" (line 95).
Comments 5: Material and methods. Ok
Response 5: Thank you.
Comments 6: Results. Table 3:
Ref Barsky et al.27, Country, should say Canada in addition to East Africa
Ref Koopman et al. 32 , revise the name of the country.
Ref Fisher et al. 34, revise the name of the country.
Ref Levine et al. 36, revise the name of the country.
Ref Ashjian et al. 37, revise the name of the country.
Ref Manusov et al. 46, revise the name of the country.
Ref Lee et al. 47, revise the name of the country.
Ref Davoudi et al. 51, revise the name of the country.
Ref Nau et al. 58, revise the name of the country.
Ref Ye et al. 59, revise the name of the country.
Response 6: Thank you for the feedback.
We have revised the name of the country based on your suggestion in Table 3.
Comments 7: Discussion. Please comment on the differences between countries and health systems. It is not the same as private vs public primary care. And telehealth /e-health technology in developing vs developed countries.
Response 7: Thank you for taking the time to write this.
We have added comment on the differences between countries and health systems and telehealth/e-health technology in developing vs developed countries in discussion part. See lines 86-100.
Comments 8: Limitation. Authors should recognize publication bias, because of the language. They only included articles in English.
Response 8: Thank you for thic comment.
We have included the publication bias in the limitation (line 106).
Comments 9: Comments on the Quality of English Language
Please correct typos, line 36 says “hypertion” I suppose it should say “hypertension”; and line 57 says “hypertensionin” I suppose it should say “hypertension in”.
Response 9: Thank you for the comment.
We have corrected the typos as suggested.

Round 2
Reviewer 1 Report
Comments and Suggestions for Authors
Again, no yellow changes in the manuscript. Please mark changes in yellow
Comments on the Quality of English LanguageRespected Editor,
again no yellow changes in manuscript. Please ask authors for marking changes in yellow.
Joanna
Author Response

(The authors gave the same response as above.)

Round 3
Reviewer 1 Report
Comments and Suggestions for Authors
Manuscript improved.